# LPS-Induced Coagulation and Neuronal Damage in a Mice Model Is Attenuated by Enoxaparin

**DOI:** 10.3390/ijms231810472

**Published:** 2022-09-09

**Authors:** Shani Berkowitz, Shany Guly Gofrit, Shay Anat Aharoni, Valery Golderman, Lamis Qassim, Zehavit Goldberg, Amir Dori, Nicola Maggio, Joab Chapman, Efrat Shavit-Stein

**Affiliations:** 1Department of Neurology, The Chaim Sheba Medical Center, Ramat Gan 52626202, Israel; 2Department of Neurology and Neurosurgery, Sackler Faculty of Medicine, Tel Aviv University, Tel Aviv 6997801, Israel; 3Goldschleger Eye Institute, Sheba Medical Center, Ramat Gan 52626202, Israel; 4Talpiot Medical Leadership Program, The Chaim Sheba Medical Center, Ramat Gan 52626202, Israel; 5Sagol School of Neuroscience, Tel Aviv University, Tel Aviv 6997801, Israel; 6Robert and Martha Harden Chair in Mental and Neurological Diseases, Sackler Faculty of Medicine, Tel Aviv University, Tel Aviv 6997801, Israel; 7Department of Physiology and Pharmacology, Sackler Faculty of Medicine, Tel Aviv University, Tel Aviv 6997801, Israel

**Keywords:** enoxaparin, neuroinflammation, thrombin, protease-activated receptor

## Abstract

Background. Due to the interactions between neuroinflammation and coagulation, the neural effects of lipopolysaccharide (LPS)-induced inflammation (1 mg/kg, intraperitoneal (IP), n = 20) and treatment with the anti-thrombotic enoxaparin (1 mg/kg, IP, 15 min, and 12 h following LPS, n = 20) were studied in C57BL/6J mice. Methods. One week after LPS injection, sensory, motor, and cognitive functions were assessed by a hot plate, rotarod, open field test (OFT), and Y-maze. Thrombin activity was measured with a fluorometric assay; hippocampal mRNA expression of coagulation and inflammation factors were measured by real-time-PCR; and serum neurofilament-light-chain (NfL), and tumor necrosis factor-α (TNF-α) were measured by a single-molecule array (Simoa) assay. Results. Reduced crossing center frequency was observed in both LPS groups in the OFT (*p* = 0.02), along with a minor motor deficit between controls and LPS indicated by the rotarod (*p* = 0.057). Increased hippocampal thrombin activity (*p* = 0.038) and protease-activated receptor 1 (PAR1) mRNA (*p* = 0.01) were measured in LPS compared to controls, but not in enoxaparin LPS-treated mice (*p* = 0.4, *p* = 0.9, respectively). Serum NfL and TNF-α levels were elevated in LPS mice (*p* < 0.05) and normalized by enoxaparin treatment. Conclusions. These results indicate that inflammation, coagulation, neuronal damage, and behavior are linked and may regulate each other, suggesting another pharmacological mechanism for intervention in neuroinflammation.

## 1. Introduction

Neuroinflammation underlies the pathogenesis of various heterogeneous neurological manifestations, including stroke [1], traumatic brain injury (TBI) [2], amyotrophic lateral sclerosis (ALS) [3], diabetic neuropathy [4], and autoimmune diseases such as multiple sclerosis (MS) [5]. In certain manifestations such as stroke and TBI, the neuroinflammatory process is secondary to the main insult [6,7], while it is the primary cause in other cases such as systemic lupus erythematosus (SLE) [8,9] and sepsis [10]. The latter may cause well-characterized changes in mental functions known as sepsis-associated encephalopathy. Long-term neurological effects following an acute and intensive inflammatory response are described in the COVID-19 sequela, known as persistent post-COVID syndrome [11], supporting a chain of events that start with inflammation and result in long-term neurologic damage. 

The coagulation system is well connected to inflammation and is evident in systemic [12] and neurological diseases [13,14]. Coagulation is initiated in inflammatory settings, and endogenous anticoagulant processes are reduced, along with the fibrinolytic system activation. Inflammatory cytokines can act as the main mediators in coagulation activation [15,16,17]. Thrombin mediates its cellular effects through proteolytic activation of protease-activated receptors (PARs), which consists of four G-protein coupled receptors. In this family, PAR1 is considered to be the main thrombin receptor [18]. Coagulation factors and their receptors are intrinsically expressed in neural tissues [19] and PARs are well-known molecular targets for their participation in the development of inflammatory disorders [20]. Previous studies have shown improvement in neuroinflammation upon inhibition of the serine protease thrombin [21,22,23]. Therefore, pharmacological intervention in the coagulation cascade should be further investigated as a potential treatment for neuroinflammation. 

Enoxaparin is a homogenous low-molecular-weight derivative of heterogenic unfractionated heparin [24], which has been in medical use for centuries [25]. Enoxaparin mostly catalyzes the creation of the anti-thrombin-factor (F) Xa complex, and, to some degree, the creation of the thrombin-anti-thrombin complex. Thus, enoxaparin inhibits two main components of the coagulation cascade [24]. Enoxaparin is indicated for the treatment and prophylaxis of hypercoagulability states. 

Aside from its obvious anti-coagulation indications, long-term administration of enoxaparin shows a beneficial neurological effect in Alzheimer’s disease (AD) mice models. Enoxaparin reduces the presence of reactive astrocytes in the vicinity of β amyloid plaques, as well as the immune activation caused by such plaques [26]. In an animal model for TBI, short-term administration of enoxaparin was reported to have a positive effect. Administering enoxaparin following TBI in rats resulted in reduced brain edema and improved cognitive function [27]. Shengjie et al. found reduced leukocytes rolling on the endothelium and improved blood–brain barrier (BBB) integrity following enoxaparin administration in TBI animals [28]. 

A possible mechanism for this beneficial effect involves the blocking of the high mobility group box 1 (HMGB1) protein, which is responsible for leukocyte recruitment. However, in this model, lung edema was reduced by enoxaparin and not by direct inhibition of HMGB1, suggesting a more complicated mechanism of action [29]. Inflammasome proteins caspase-1 and interleukin 1β were elevated following TBI in both the brain and lung. Enoxaparin administration reduced their levels and improved the lung injury which often accompanies TBI [30]. 

Previous research has described the beneficial effects of enoxaparin in neurological damage associated with inflammation secondary to the main insult in animal models but has not yet been explored in the setting of direct inflammation. Systemic injection of lipopolysaccharide (LPS) is used as a model for systemic inflammation. Systemic LPS injection affects the nervous system both immediately following administration [31], and chronically, months after LPS injection [32]. Well-established temporal effects in this model are characterized by an early increase of systemic pro-inflammatory mediators (starting 2 h post-injection) [32,33], while long-lasting pro-inflammatory factors are seen at later time points in the brain (for weeks and months) [32,34]. Mild cognitive effects, reduced locomotion, and increased anxiety can be seen as soon as 2–7 days [35] and up to weeks after systemic injection [34,36]. Progression of inflammation may cause inflamed tissue to exhibit hyperalgesia [37,38]. Previously, LPS has been shown to increase coagulation factors and their cellular receptors locally in the hippocampus [21]. The use of enoxaparin in an LPS model may shed light on our understanding of the mechanism behind the neural deficits seen following LPS injection.

The specific effect of enoxaparin in the context of primary inflammation, and whether the modulation of the coagulation cascade affects inflammation, neuronal function, or both, remains an open question. It is still unclear which of these effects predominate. In the present study, we utilized a classic LPS model that initiates inflammation and found that this also triggers coagulation, particularly in the brain. We used enoxaparin to counter the deleterious effects of the LPS-induced neuroinflammation and assessed whether it primarily affects coagulation or inflammation in this model. We hypothesized that enoxaparin treatment will significantly attenuate LPS-induced coagulation and neuronal damage. Our results indicate that the LPS neuroinflammatory model induces coagulation in the brain and neuronal destruction as evident by increased neurofilament light chain (NfL) levels in the serum. For the first time, we have found that enoxaparin reduces this cellular damage predominantly by its effect on coagulation without a major effect on brain inflammation. 

## 2. Results

### 2.1. General Health of the LPS Mice

Both groups of animals with LPS-induced neuroinflammation showed significantly lower body weight during days 1–3 (ANOVA, *p* < 0.0001, Figure 1A). This was compatible with the known sickness behavior in this period owing to an inflammatory effect. By day 4, weight was not significantly different between LPS and controls (21.79 ± 0.37 and 22.75 ± 0.42 gr, respectively, *p* = 0.16) or between enoxaparin-treated LPS and controls (21.98 ± 0.37 gr, *p* = 0.29).

### 2.2. Motor Functions

Motor function was assessed by the rotarod test on day 5. A trend for motor function deficits between the controls, LPS, and enoxaparin-treated LPS was seen (59.91 ± 0.08, 56.22 ± 1.80, 57.38 ± 0.88, seconds, respectively, ANOVA, *p* = 0.057, Figure 1B). This suggested deficit in motor abilities led us to carefully assess the motor performance of each mouse in further behavioral and cognitive tests.

### 2.3. Heat Sensitivity Function

No significant changes in heat response were noted between control, LPS, and enoxaparin-treated LPS on day 6 (18.6 ± 1.7, 18.6 ± 1.8, 20.6 ± 1.5 s, respectively, ANOVA, *p* = 0.6, Figure 1C).

### 2.4. Open Field Test

The open field test measures locomotor capabilities and anxiety-like behavior [39]. The LPS and enoxaparin-treated LPS tended to move a shorter distance compared to the controls, but the difference did not reach significance (1655 ± 95.72, 1668 ± 87.36, 1907 ± 81.69 cm, respectively, ANOVA, *p* = 0.08, Figure 2A), supporting a mild motor impairment in the LPS groups on day 5. The crossing center frequency of the LPS and enoxaparin-treated LPS groups decreased compared to controls (12.60 ± 1.32, 12.35 ± 1.2, 17 ± 1.4, *p* = 0.02, Figure 2B). These results support increased anxiety-like behavior in the LPS groups, which was not improved by the enoxaparin treatment.

### 2.5. Y-Maze

The Y-maze was employed to assess spatial memory. The time spent and distance traveled in the familiar and novel arms were used to index memory. The LPS and enoxaparin-treated LPS moved a shorter distance compared to the controls (965.3 ± 44.54, 865.3 ± 29.53, 1053 ± 41.85 cm, respectively, ANOVA, *p* < 0.01, Figure 2C). The recognition index was similar for all three groups (0.70 ± 0.02, 0.67 ± 0.02, 0.72 ± 0.02, for LPS, LPS enoxaparin, and control, respectively, ANOVA, *p* = 0.46, Figure 2D).

### 2.6. Thrombin Activity

Thrombin activity was assessed one week after LPS injection in three different neural-associated tissues. The hippocampus was studied as a representative central neural tissue which is also associated with cognitive function. The sciatic nerve was studied as a representative peripheral neural tissue containing a variety of fibers partaking in neuronal conduction. A recent report by our group showed correlation between thrombin activity and skin innervation [40]. Therefore, thrombin activity in the skin was measured in the current report. A significant difference was seen in the hippocampus between the control and the LPS group (0.03 ± 0.009 and 0.06 ± 0.015 mU/mg, respectively, Mann–Whitney, *p* = 0.04, Figure 3A). This difference was not detected in the enoxaparin-treated LPS (0.03 ± 0.014 mU/mg, *t*-test, *p* = 0.4). Enoxaparin-treated LPS mice had lower hippocampal thrombin activity compared to LPS, a decrease which did not reach significance (0.03 ± 0.014, 0.06 ± 0.015 mU/mg, respectively, Mann–Whitney, *p* = 0.16). Measurements of sciatic thrombin activity at the same time point showed a non-significant elevation in the LPS and enoxaparin-treated LPS compared to controls (1.04 ± 0.16, 0.98 ± 0.14, 0.73 ± 0.14, mU/mg, respectively, ANOVA, *p* = 0.3, Figure 3B). Measurements of skin thrombin activity showed no significant differences between the LPS and enoxaparin-treated LPS groups, compared to controls (3.25 ± 0.78, 5.64 ± 1.46, and 3.53 ± 0.86, mU/mg, respectively, ANOVA, *p* = 0.25, Figure 3C).

### 2.7. Hippocampal Gene Expression of Coagulation and Inflammatory Factors

Hippocampal coagulation gene expression in the LPS group was altered one week after injection. LPS mice had significantly elevated PAR1 expression compared to controls (1.36 ± 0.12, 1.0 ± 0.06, respectively, *p* = 0.01, Figure 4A). Enoxaparin-treated LPS had normalized levels of PAR1 expression compared to LPS (1.03 ± 0.05, 1.36 ± 0.12, respectively, *p* = 0.03, Figure 4A). Prothrombin levels showed a trend towards a decrease in the LPS group compared to controls and enoxaparin-treated LPS (0.78 ± 0.07, 1 ± 0.08, 0.99 ± 0.11, respectively, *p* = 0.1, Figure 4B). FX levels did not differ between LPS compared to controls and enoxaparin-treated LPS (0.87 ± 0.08, 1 ± 0.06, 1.06 ± 0.23, respectively, *p* = 0.4, Figure 4C). IL1-β was not significantly different between LPS compared to controls and enoxaparin-treated LPS (1.2 ± 0.07, 1 ± 0.07, 1.06 ± 0.1, respectively, *p* = 0.3, Figure 4D). TNF-α was significantly increased in the LPS group compared to controls (1.57 ± 0.18 and 1 ± 0.05, respectively, Kruskal–Wallis, *p* = 0.02). TNF-α expression in the enoxaparin-treated LPS group was not affected by treatment and was similar to LPS (1.6 ± 0.27, *p* = 0.99, Figure 4E).

### 2.8. Serum Markers for Axonal Damage and Inflammation

Serum NfL levels were measured in all groups one week following LPS injection. Significant changes were found between the three groups (*p* = 0.04, Kruskal–Wallis). NfL levels were significantly higher in the LPS mice compared to controls (319.8 ± 95, 89.1 ± 11.5, pg/mL, respectively, *p* = 0.01, Mann–Whitney, Figure 5A), indicating neuronal damage. NfL levels in the enoxaparin-treated LPS were similar to control NfL levels (96.1 ± 29.5 pg/mL, *p* > 0.99). Additionally, a significant difference between the enoxaparin-treated LPS and the LPS groups was seen (*p* = 0.02, Mann–Whitney), suggesting a neuroprotective effect.

Serum TNF-α levels were elevated significantly in the LPS group compared to the controls (81.43 ± 19.09, 36.45 ± 2.20, pg/mL, respectively, *p* < 0.01). This significant increase was prevented in the enoxaparin-treated LPS group compared to controls (46.08 ± 6.66 pg/mL, *p* = 0.55, Figure 5B).

## 3. Discussion

In this study, we investigated the effect of enoxaparin treatment on behavioral and biochemical parameters in an inflammatory LPS animal model. Our results indicate impaired locomotive abilities and some behavioral changes, together with an elevation of hippocampal thrombin activity, as well as an elevation of intrinsic hippocampal PAR1 and expression of inflammatory factors. These results, with the elevation of serum NfL and TNF-α levels, indicate neuronal damage and inflammation, respectively. Enoxaparin treatment attenuated the increase in thrombin activity and PAR1 expression in the hippocampus as well as serum NfL and TNF-α elevation, supporting a neuroinflammatory-protective effect.

Impaired locomotion is a part of sickness behavior in response to infection. Other characterizations include reduced appetite and depressive mood [41], which manifested in the present work as reduced weight in both the LPS groups, followed by normalization. These characteristics are replicated in the LPS animal model [42,43], along with cognitive impairment due to neuroinflammation [35], which can be seen as a reduction in the crossing center frequency in the open field tests. Mildly reduced locomotive abilities following LPS injection are seen in the present study as decreased distance in the open field and Y maze tests. However, the less sensitive rotarod test [44] did not detect a significant motor impairment. The rotarod test was performed at a single speed; therefore, it is possible that a different speed would have revealed motor deficits. The combination of the relatively normal rotarod test and the reduced distance seen in the behavioral tests support the finding of minor motor impairment in the presence of behavioral deficits such as increased anxiety. Our data indicated minor motor deficits that may represent specific temporal aspects of LPS-induced effects. Sickness behavior is evident in several measures. Some studies support sickness behavior returning to baseline after 24 h [45]. Cognitive changes are detected in the first days following injection and improve over time [35]. This narrow time window, which allows for conducting behavioral cognitive tests in the relative absence of sickness behavior, guided the design of the current experimental timeline. In our study, the weight returned to normal on day 4, and motor function seen in the Y-maze tests did not normalize until 6 days after LPS injection. Pain may be another possible explanation for the reduced locomotion; however, we did not detect increased heat sensitivity. Although relatively small, the presence of a motor impairment complicates drawing concrete conclusions from the behavioral tests, which measure anxiety and spatial memory based on locomotion. Previous studies supporting anxiety modifications in animal models of antiphospholipid syndrome suggest an interaction between inflammation, coagulation, and anxiety [46,47,48]. Future studies may need to focus on cognitive evaluation assessed at different points in time or should include observational studies on the effect of infection on NfL in human subjects in relevant clinical scenarios in which enoxaparin is indicated as part of routine treatment.

The site of LPS-induced damage, as well as the site of enoxaparin involvement, remains an open question. A previous study supported relatively minor changes in BBB permeability following low concentrations (0.3 mg/kg) of systemic LPS injection, with significant penetration in higher concentrations (3 mg/kg) [49]. In the present study, we used a dosage of 1 mg/kg, expected to cause some degree of BBB disruption. LPS injection caused an increase in brain mRNA TNF-α expression, both in the LPS and the LPS enoxaparin group, further supporting the effect of LPS in the CNS. Thrombin was previously shown to activate C6 glioma cells in vitro, resulting in the expression of TNF-α [50]. Evidence supports the activation of microglia via the PAR1 pathway [51]. Activation of cultured microglia results in the release of proinflammatory cytokines including TNF-α to the medium, which in turn reduces dopaminergic neurons [52]. Interestingly, serum TNF-α levels were significantly increased in the LPS group but not in the LPS enoxaparin group, suggesting an interaction between enoxaparin and systemic inflammation. Our results are in line with previous research indicating that enoxaparin has additional anti-inflammatory effects beyond coagulation [53]. However, hippocampal TNF-α presence was assessed by mRNA expression, while serum TNF-α was evaluated by protein concentration, which may reflect not only spatial but also temporal differential effects. Protein analysis of TNF-α levels in the hippocampus at the same time point would answer this question, but the measurement is technically challenging due to extremely low levels of the cytokine. Normalization of weight, which occurred from day 4 to day 7, may represent the subsiding of the peripheral inflammatory response, supporting a time lag between peripheral and central inflammatory responses. The source of neuro-inflammation, either central with a later peripheral response, or having concomitant central and peripheral components, remains to be evaluated. Novel and highly sensitive methods for measurements of serum NfL have provided an accessible surrogate for the evaluation of neuronal damage. Elevated levels of NfL have been seen in many neurological manifestations that include neurodegenerative and neuroinflammatory processes such as multiple sclerosis, ALS, TBI, and dementia [54]. Using the precise Simoa instrument, we were able to detect key changes in the levels of NfL. Serum NfL levels were significantly elevated in LPS mice, and this elevation was blocked by enoxaparin, suggesting a protective effect. In addition, lack of heat sensitivity, as was demonstrated by the hot plate test together with the non-significant changes in thrombin activity in the sciatic nerve and skin, further strengthens a CNS origin but does not rule out peripheral involvement. Measuring both peripheral inflammatory markers as well as markers for peripheral neuronal damage may aid in the localization of the injury and will take place in future human-based research. Whether the central inflammation is a sequela of the peripheral inflammation or a direct response to the penetration of LPS, it is now clear that the neuronal coagulation system is an important mediator and a site for intervention.

Enoxaparin significantly reduced NfL levels, supporting its neuroprotective effect. However, the site of this protective effect is not clear. One possibility is BBB integrity. Under pathological conditions, high concentrations of enoxaparin slowly crossed the monolayer of the BBB [55], suggesting some degree of effect directly in the CNS. However, enoxaparin inhibits cytokine release from mononuclear cells in the peripheral blood [56], raising the possibility that enoxaparin exerts its protective effect by reducing the inflammatory response outside the CNS. Previous work in a TBI mouse model supports a combination of these two options; enoxaparin treatment reduced leukocyte recruitment as well as improved BBB integrity post-TBI [28].

Aside from the intervention site, the mechanism by which enoxaparin protects against neuronal damage remains a matter of debate. Enoxaparin potentiates anti-thrombin to bind mostly FXa, and, to some extent, thrombin [57]. The major inhibitory effect on the distinct phase of thrombin generation, as well as on endogenous thrombin potential, is due to anti-FIIa activity [58]. We found that enoxaparin reduces hippocampal mRNA expression of thrombin receptor PAR1, but not prothrombin or FX, suggesting an effect on transcription. The significantly increased PAR1 expression in the LPS group only suggests an inflammatory-induced change that is intrinsic to the brain, as part of the systemic inflammatory response. This result may point toward a feedback loop caused by the upstream inhibition of FXa by enoxaparin followed by reduced thrombin activity, and subsequently, the elevation of PAR1 protein levels resulting in reduced PAR1 production. Direct thrombin inhibition by enoxaparin may result in a reduction of PAR1 activation accompanied by a reduction of PAR1 mRNA expression, perhaps as a feedback loop. Since activation of PAR1 by thrombin supports inflammation [59], its inhibition by enoxaparin may be the mechanism by which it provides neuronal protection. However, other mechanisms involving its effect on FXa may be involved as well. Further study is needed to characterize the enoxaparin effect in the CNS. Evaluation of other previously described neuromodulatory effects in the PAR1 pathway [60] may shed light upon this mechanism.

Our study has several limitations. Although this study did not include a control group treated with enoxaparin only, we assume that its effects are due to the interaction between enoxaparin and inflammation during increased BBB disruption. This is supported by a previous study indicating that enoxaparin treatment in healthy wild-type mice did not affect brain intrinsic inflammation and behavior [61]. As mentioned above, evaluation of peripheral inflammatory markers is difficult in mice, calling for future research in other models. Using novel in vivo thrombin activity probes [62,63] may shed light upon other physiological aspects in this model. The dosage of enoxaparin treatment is limited by extensive bleeding as a side effect. There is a need for a selected intervention to neutralize other variables in the PAR1 pathway, using small, highly specific molecules.

## 4. Materials and Methods

### 4.1. Model Establishment and Treatment Protocol

All experiments were approved by the Sheba Medical Center Animal Welfare Committee (1295/21/ANIM) and appropriate measures to avert pain and suffering to the animals were taken. All animals were maintained in a controlled animal facility at 18–22 °C and 40–60% humidity, with a photoperiod of 12 h dark/12 h light, and were treated according to the ARRIVE guidelines. During the experiment, animals were allowed free access to water and food. Animals were weighed daily for evaluation of general health.

The 8-week-old male C57BL/6J mice were purchased from Envigo Laboratories, Israel. The experiment was repeated twice, each time with 10 animals per group (30 mice in round 1 and 30 mice in round 2). In each repetition, the animals were allocated to three groups: control animals were treated with saline injection (n = 10); LPS animals were injected intraperitoneally (IP) with LPS (Escherichia coli 0111:B4, Sigma L4130, 1 mg/kg, diluted in saline) at day 0 (n = 10), and LPS enoxaparin animals were injected with LPS as described above, along with two IP enoxaparin injections (1 mg/kg, Sanofi, Paris, France) at 15 min and 12 h following LPS administration (n = 10). Coagulation and inflammation factor levels and activity were assessed in 10 animals from each group. Animals were sacrificed on day 7 by pentobarbital injection (0.8 mg/kg) for measurements of thrombin activity, hippocampal mRNA levels of coagulation and inflammation factors, and evaluation of NfL and inflammatory marker levels in the serum. Figure 6 describes the experiment timeline.

### 4.2. Rotarod

Motor performance was assessed utilizing a rotarod test on day 5 (n = 20 in each group). Mice were pre-trained prior to LPS treatment to run on the rod, which rotated at a fixed speed of 19 revolutions per minute. Mice were allowed to run for up to 60 s on each trial, or until they fell off. The mean of the three consecutive trials was recorded for each animal. Mice that fell during the first 10 s of the first trial were returned to the rod for continued assessment. One animal was excluded from the control group due to jumping off the rod repeatedly.

### 4.3. Hot Plate

On day 6, hyper/hypoalgesia was evaluated using the hot plate test. Mice were placed in an acrylic glass cylinder on a digital heated stage maintained at 51 ± 0.1 °C. Time to heat response indicated by hind paw licking, shaking or jumping was measured. A maximum on-plate time was set to 30 s to prevent skin injury. The hot plate was performed in the first repetition of the experiment.

### 4.4. Behavioral Tests

All behavioral tests were conducted by a researcher who was blinded to the group treatment allocation. The experiment was conducted during the daytime between 8 AM and 4 PM at 22 °C, and in the lighting of 240 lux. The animals were all allowed to adjust to the behavioral acquisition settings prior to testing.

### 4.5. Open Field

The open field test was conducted on day 5 (n = 20 in each group) in a square apparatus (39.5 × 39.5 × 30 cm) made of black acrylic material. At the beginning of the test, each mouse was placed in the center of the apparatus and allowed to move freely for a single exploration trial. Each mouse was trace-recorded with a ceiling-mounted video camera (Tracker VP200; HVS Image, Hampton, England) for 5 min (with a delay of 30 s from placement at the center of the arena, to avoid noise disturbance). The floor and walls of the field were cleaned thoroughly with ethanol and air-dried after each trial to remove olfactory cues. Data analysis was employed using a tracking system (Ethovision by Noldus, NL), conducted by a researcher who was blinded to the experimental groups. The field image was divided into 16 equally sized squares and defined as follows: Center: 4 inner squares, Corners: 4 corner squares. One mouse was excluded from the control group due to a significantly decreased moving distance (a difference of ten standard errors from the mean).

### 4.6. Y-Maze

Evaluation of spatial memory using the Y-maze was conducted on day 6 (n = 20 in each group) [64]. The Y-shaped apertures consisted of three arms. Spatial cues included vertical tape positioned at the entrance to the familiar arm and a triangle at the entrance to the novel arm. Mice were placed in an arm of the Y-maze (entrance arm) with one of the arms blocked off (novel arm). Mice were allowed to explore the start arm and remaining arm (familiar arm) for 5 min. After a 1 min break, mice were placed back in the entrance arm and allowed to explore all arms freely for 2 min. The time spent in each arm of the maze was recorded. Recognition capabilities were assessed by calculating the recognition index defined as [duration time in novel arm/ (duration time in novel arm + duration time in the familiar arm)]. One mouse was excluded from the control group due to significantly reduced performance (a difference of ten standard errors from the mean).

### 4.7. Thrombin Activity Assay

Thrombin enzymatic activity was measured using a fluorometric assay based on the cleavage rate of the synthetic substrate Boc-Asp(OBzl)-ProArg-AMC (I-1560; Bachem, Bubendorf, Switzerland) and defined by the linear slope of the fluorescence intensity versus time, as previously described [21,65,66,67,68]. One week following LPS injection, mice (n = 20 in each group) were anesthetized with pentobarbital. Based on our previous experience, in some disease states, the pathology itself limits perfusion efficiency [69]. In setting up the assay there was no significant difference in the levels of hippocampus thrombin activity measured in a group of 6 perfused compared to 4 non-perfused animals (Appendix A). In order to perform efficient perfusion and avoid thrombus generation during the procedure, heparin is routinely used in the first washing step. In the present study, the use of heparin is unacceptable since it significantly affects thrombin activity and this would modify and bias the measurements. Since we studied a disease model that potentially affects brain vessel thrombosis and permeability, we did not perfuse the animals. Brains were removed for hippocampus dissection. The right half of the hippocampus and one sciatic nerve were collected for the thrombin activity assay. A 3 mm punch glabrous (hairless) skin biopsy was obtained from one hindfoot. We used whole tissue preparation rather than homogenates since previous studies indicate a vast number of protease inhibitors are released upon homogenization and inhibit protease activity [70,71]. The inhibitory activity released by homogenization seems to be intrinsic to the tissue since it is not affected by perfusion [71]. Hippocampal tissue, unsheathed sciatic nerves, and skin biopsies were placed into a 96-well black microplate (Nunc, Roskilde, Denmark) containing Tris buffer (50 mM Tris HCl, pH 8, 150 mM NaCl, 1 mM CaCl_2_). Measurements were carried out using a microplate reader (Tecan; Infinite F Nano+; Männedorf, Switzerland) with excitation and emission filters of 360 ± 35 and 460 ± 35 nm, respectively. Measurements were conducted continuously in the presence of the tissue using the heat (37 °C), shake, and top-reading modes (both excitation and emission applied vertically) with a high sampling rate and 25 flashes per reading, and 25 µs integration time. This high sensitivity and sampling rate enables us to reduce the tissue interference potential with the reading. In cases of tissue interference noise, this was manually excluded without affecting the general slope. In addition, all wells contained similar tissue, and all slopes were compared to each other. Preliminary experiments performed on 10 hippocampus samples found that when the samples are moved to new wells, a small but measurable activity is retained in the sample (Appendix A). These experiments also demonstrate that the tissue has a stable level of auto-fluorescence which only very rarely (1% of measurements) affects the read. This method enabled performing the assay with less time delay following extraction of the tissue and enabled us to detect a linear increase in fluorescence signal representing the enzymatic activity. Furthermore, not all thrombin activity in the tissue is necessarily soluble. In cell culture experiments performed in similar wells, most of the thrombin activity is found in the medium but a significant proportion is measured on the cells themselves [70]. Performing the assay in the presence of the tissue enables reliable measurement of both potential pools of thrombin activity.

Hippocampal, sciatic, and skin tissues were homogenized at maximal speed in a radioimmunoprecipitation assay (RIPA) buffer (50 mM Tris HCl, pH 7.6, 150 mM NaCl, 1% NP-40, 0.5% Sodium Deoxycholate, and 0.1% SDS) supplied with commercial Protease Inhibitor Cocktail (P-2714, Sigma-Aldrich, Saint Louis, MO, USA) (200, 40, 200 µL, respectively) with a bullet blender homogenizer (BB*24B, Next Advance, Troy, NY, USA) (5, 5, 2 min, respectively). Homogenates were incubated on ice for 10 min and centrifuged (13,000× *g*, 10 min) at 4 °C. Supernatants were collected and immediately placed on ice for protein concentration measurement. A bicinchoninic acid (BCA) kit (QPRO-BCA kit, Cyanagen, PRTD1,0500) was used to determine protein concentration. Reported values are normalized to the protein concentration of each sample (±SEM).

### 4.8. Coagulation and Inflammation Gene Expression in the Hippocampus

Prior to the harvest, the animals were anesthetized with pentobarbital (0.8 mg/kg). The brains were removed, and the hippocampi were dissected. Hippocampal mRNA was extracted by the addition of lysis buffer according to the Bio-Rad Aurum 732–6820 kit instructions (Bio-Rad Laboratories, Hercules, CA, USA). One microgram of total RNA was used for reverse transcription using a high-capacity cDNA reverse transcription kit (Applied Biosystems). The quantitative real-time polymerase chain reaction was performed on the StepOne™ Real-Time PCR System (Applied Biosystems, Rhenium, Israel) using Fast SYBR Green Master (ROX) (Applied Biosystems). Hypoxanthine guanine phosphoribosyltransferase (HPRT) served as a reference gene in this analysis (primer list). A standard amplification program was used: 1 cycle of 95 °C for 20 s (s) and 40 cycles of 95 °C for 3 s and 60 °C for 30 s. The primers used in this analysis are listed in Table 1. The results were normalized to reference gene expression within the same cDNA sample and were calculated using the ΔCt method with results reported as fold changes relative to the control animal’s hippocampus and reported as mean ± SE.

### 4.9. Neurofilament Light Chain and Tumor Necrosis Factor-α Measurement in the Serum

Mice were anesthetized and blood samples were collected. The collected blood was incubated at room temperature for 30 min to enable clot formation and centrifuged at 1500 g for 10 min. The serum was frozen and stored at −80 °C until analysis. For NfL measurement, sera were initially diluted at 1:1000. When an obtained concentration was higher than 500 pg/mL, an additional dilution of 1:5000 was further tested. For tumor necrosis factor α (TNF-α), sera samples were diluted 1:8. NfL concentrations were measured in duplicates by a single molecule array (Simoa) assay (Quanterix, Boston, MA, USA) employing commercial kits (NF-light Advantage kit and mouse TNF-α discovery kit for HD-1/HD-X adjusted for SR-X, UmanDiagnostics Umea, Sweden), using a bead-conjugated immunocomplex. The immunocomplex was applied to a multi-well array designed to enable imaging of every single bead. The average number of enzymes per bead (AEB) of each sample was interpolated onto the calibrator curve constructed by AEB measurements on commercial NfL and TNF-α (UmanDiagnostics), serially diluted in an assay diluent. Samples were analyzed using one batch of reagents. Animal treatment information was blinded for the investigator performing the analysis.

### 4.10. Statistical Analysis

Statistical analyses and graphs were conducted and made using GraphPad Prism (version 8.00 for Windows, GraphPad Software, La Jolla, CA, USA, www.graphpad.com). Unpaired t-tests, one-way ANOVA, and two-way ANOVA followed by a post hoc test were applied to normally distributed data sets. One-way ANOVA was followed by either Dunnett’s or Tukey’s post hoc analyses. Two-way ANOVA was followed by Sidak’s post hoc analysis. Flowing normality evaluation by D’Agostino–Pearson, Mann–Whitney, or Kruskal–Wallis tests were applied on non-normal distributed data sets. Results are expressed as mean ±SEM; *p* values < 0.05 were considered significant.

## 5. Conclusions

In conclusion, the present study highlights the involvement of the thrombin PAR1 pathway in the crosstalk between inflammation and neural damage, supported by a protective effect of enoxaparin in the CNS during systemic inflammation. Further study is needed to improve and modulate PAR1 intervention during inflammation.

## Figures and Tables

**Figure 1 ijms-23-10472-f001:**
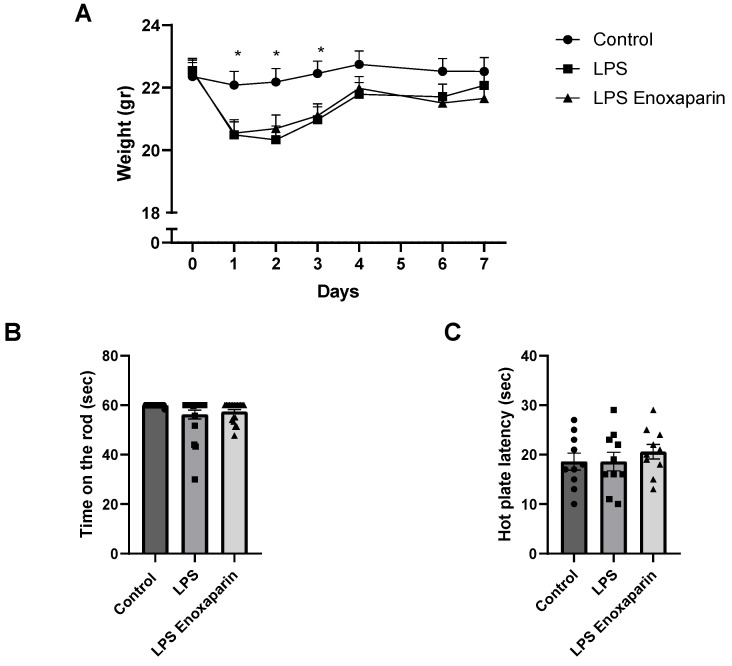
General health, motor deficits, and heat sensitivity: (**A**) LPS animals had significantly lower body weight until day 3, followed by improved body weight which normalized one week after injection (n = 20 for each group). (**B**) A trend toward motor deficits was observed between the LPS animals and the controls (control: n = 19, LPS: n = 20, LPS enoxaparin: n = 20). (**C**) No heat sensitivity changes were found between control, LPS, and enoxaparin-treated LPS (n = 10 for each group). * *p* < 0.05. LPS—lipopolysaccharide. Circles—control group; squares—LPS group; triangles—enoxaparin-treated LPS group.

**Figure 2 ijms-23-10472-f002:**
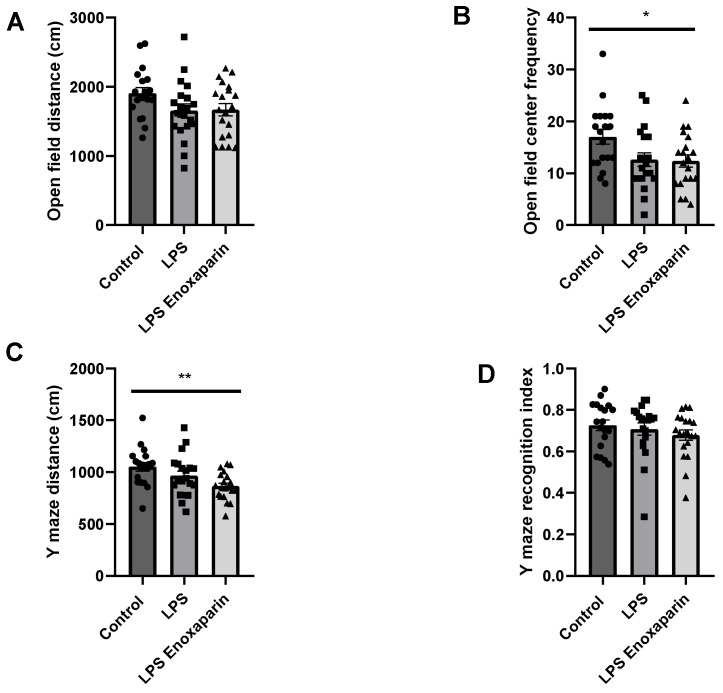
Cognitive deficits: (**A**) Both LPS and enoxaparin-treated LPS tended to travel shorter distances compared to control and crossed the center fewer times compared to control animals (**B**). (**C**) LPS animals moved significantly shorter distances in the Y-maze compared to control. (**D**) Recognition index was similar in all groups (open field: control = 19, LPS = 20, LPS enoxaparin = 20. Y-maze: control = 19, LPS = 20, LPS enoxaparin = 20). * *p* < 0.05, ** *p* < 0.01. LPS—lipopolysaccharide. Circles—control group; squares—LPS group; triangles—enoxaparin-treated LPS group.

**Figure 3 ijms-23-10472-f003:**
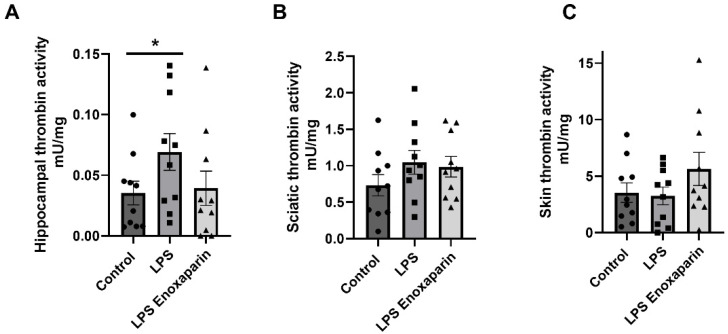
Thrombin activity levels in the hippocampus, sciatic nerve, and skin: (**A**) Thrombin activity levels were significantly elevated in the LPS group compared to controls in the hippocampus. Enoxaparin treatment prevented hippocampal thrombin activity elevation. (**B**) Thrombin activity in the sciatic nerve showed a non-significant increase. Enoxaparin treatment was not associated with a significant reduction. (**C**) No significant differences regarding skin thrombin were seen between any of the three groups (n = 10 for each group). * *p* < 0.05. LPS—lipopolysaccharide. Circles—control group; squares—LPS group; triangles—enoxaparin-treated LPS group.

**Figure 4 ijms-23-10472-f004:**
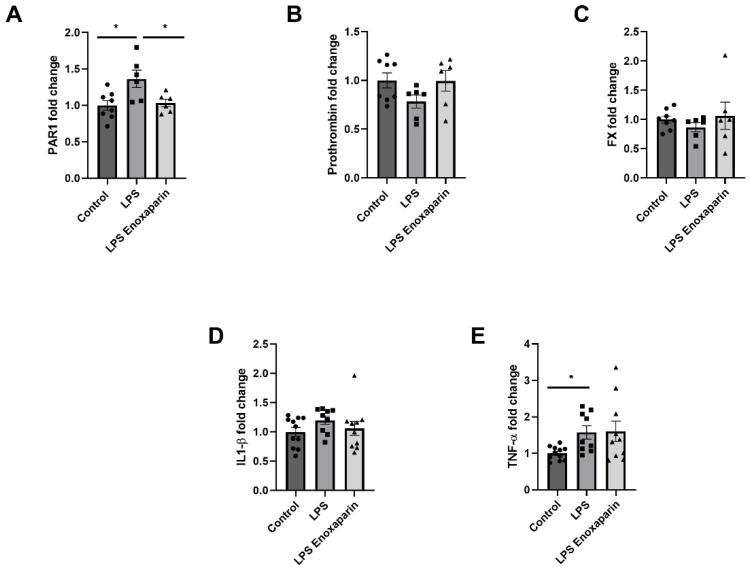
Coagulation and inflammation mRNA expression modification in the hippocampus of LPS animals one week after induction of neuro-inflammation: (**A**) PAR1 levels increased significantly in the LPS group compared to controls, with a significant reduction with enoxaparin treatment. No significant effect was measured in prothrombin (**B**) or FX (**C**) relative expression. Levels of inflammatory markers IL1-β (**D**) and TNF-α (**E**) were elevated following LPS injection and insignificantly reduced with enoxaparin treatment (coagulation markers: control = 8, LPS = 6, LPS enoxaparin = 6. Inflammatory markers: control = 11, LPS = 9, LPS enoxaparin = 10). LPS—lipopolysaccharide, PAR1—protease-activated receptor, FX—factor X, IL1-β—interleukin 1 β, TNF-α—tumor necrosis factor α. * *p* < 0.05. Circles—control group; squares—LPS group; triangles—enoxaparin-treated LPS group.

**Figure 5 ijms-23-10472-f005:**
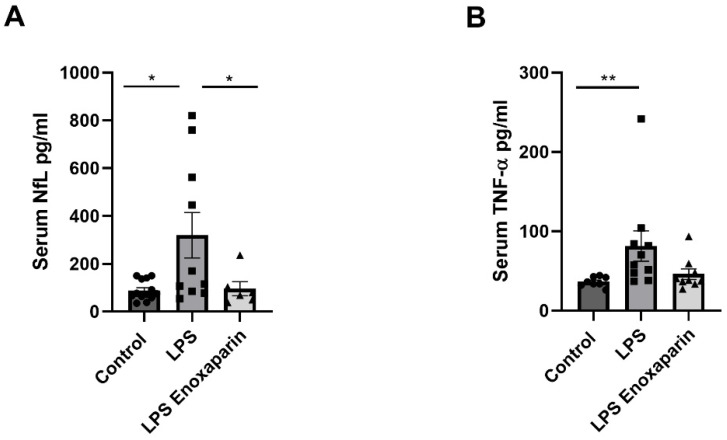
Serum NfL and TNF-α protein levels: Serum NfL (**A**) and TNF-α (**B**) levels were significantly higher in the LPS mice one week following injection compared to controls. Treatment with enoxaparin prevented the serum NfL and TNF-α elevation (NfL: control = 13, LPS = 10, LPS enoxaparin = 6. TNF-α: control = 8, LPS = 10, LPS enoxaparin = 9). NfL—neurofilament light chain; TNF-α—tumor necrosis factor α; LPS—lipopolysaccharide. * *p* < 0.05, ** *p* < 0.01. Circles—control group; squares—LPS group; triangles—enoxaparin-treated LPS group.

**Figure 6 ijms-23-10472-f006:**
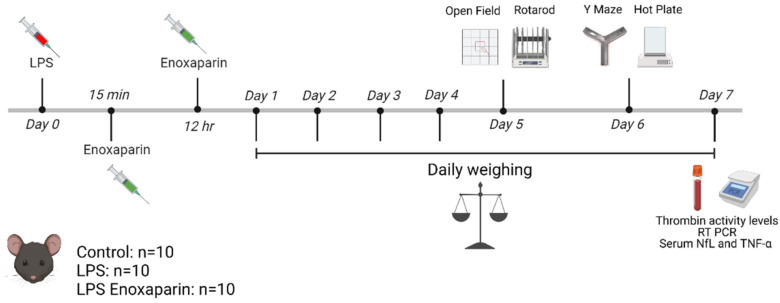
Timeline: The experiment was repeated twice. LPS was induced in 20 animals on Day 0. 10 animals in the LPS group were treated with two injections of enoxaparin on the same day. Motor performance was assessed by the rotarod on day 5. Cognitive parameters were assessed on days 5 and 6 by the open field test and Y-maze, respectively, and sensory evaluation was conducted on day 6 by the hot plate test. Animals were sacrificed on day 7 for thrombin activity, RT-PCR, and serum NfL and TNF-α quantifications. Animals were weighed daily. LPS—lipopolysaccharide, RT PCR—real time polymerase chain reaction, NfL—neurofilament light chain, TNF-α—tumor necrosis factor α. Created with BioRender.com.

**Table 1 ijms-23-10472-t001:** List of primers used to assess coagulation and inflammation mRNA expression.

Gene	Forward	Reverse
PAR1	GCCTCCATCATGCTCATGAC	AAAGCAGACGATGAAGATGCA
PT	CCGAAAGGGCAACCTAGAGC	GGCCCAGAACACGTCTGTG
FX	GTGGCCGGGAATGCAA	AACCCTTCATTGTCTTCGTTAATGA
TNF-α	GACCCTCACACTCAGATCATCTTCT	CCTCCACTTGGTGGTTTGCT
IL1-β	CTGGTGTGTGACGTTCCCATTA	CCGACAGCACGAGGCTTT
HPRT	GATTAGCGATGATGAACCAGGTT	CCTCCCATCTCCTTCATGA CA

PAR1—protease-activated receptor 1; PT—prothrombin; FX—factor X; TNF-α—tumor necrosis factor α; IL1-β—interleukin 1β; HPRT—hypoxanthine guanine phosphoribosyltransferase.

## Data Availability

The datasets used and/or analyzed during the current study are available from the corresponding author on reasonable request.

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
