# Peer review of "LPS-Induced Coagulation and Neuronal Damage in a Mice Model Is Attenuated by Enoxaparin"

_ijms, 2022, doi:10.3390/ijms231810472_

Round 1

Reviewer 1 Report (Previous Reviewer 1)

The submitted manuscript describes the cross-talk between systemic and CNS inflammation and coagulation, as well as neuroprotective and anti-inflammatory effect of anti-thrombotic agent, enoxaparin. The learning/memory tests revealed impairments triggered by systemically administered LPS, but no apparent effect of enoxaparin. However, Authors observed significant drop in LPS-induced neurofilament light chain (intermediate filament specific for neurons) release in enoxaparin treated animals, suggesting that inhibition of coagulation cascade is indeed neuroprotective. Moreover, LPS triggered TNFa mRNA expression in hippocampus, but the observed increase wasn’t inhibited by enoxaparin. On the other hand, TNFa protein level in serum was lower when enoxaparin was co-applied. Thrombin activity, as measured in whole hippocampus, was inhibited by enoxaparin. Interestingly, enoxaparin doesn’t inhibit thrombin activity in sciatic nerve and skin fragment, although one may expect its concentration and thus effect would be stronger in periphery than in brain. The interesting, but also very complex picture of interplay between systemic and local CNS coagulation and inflammation is drawn, and the presented work proves that studying effects of systemic inflammatory stimuli on neuroinflammation and coagulation in CNS is indeed very challenging.

Minor points.

1.       Line 288: “High levels of NfL measured in the serum of LPS mice may originate from the CNS, peripheral nerves, or both. However, very high levels of serum neurofilament support central neuronal damage with blood spillage  [52,53]”. Do Authors claim that neurofilament release upon LPS injection is massive, thus it must be released from neurons dying in the brain? To prove this, brain damage in the described model should be more thoroughly studied, showing for instance neurons dying in the hippocampus. Since this kind of evidence is not provided, the statements like the one above need to be avoided.

2.       Line 192: “TNF-α was significantly increased in the LPS group compared to controls (1.57±0.18 and 1±0.05, respectively, Kruskal-Wallis, p=0.02). TNF-α expression in the enoxaparin treated LPS group was not affected by treatment and was similar to LPS and control (1.6±0.18, 1.6±0.27, p=0.99, 0.07 respectively, Figure 4E)” – here TNFa levels were similar in LPS group (1.57 or 1.6?) and enoxaparin and LPS group, but not the control. Please correct.

3.       Was the effect of TNFa on coagulation and in reverse, effect of thrombin on TNFa expression, previously studied, especially in cells like neurons, astrocytes or microglia in any in vitro system? How do they affect each other?

Author Response

Reviewer 1

The submitted manuscript describes the cross-talk between systemic and CNS inflammation and coagulation, as well as neuroprotective and anti-inflammatory effect of anti-thrombotic agent, enoxaparin. The learning/memory tests revealed impairments triggered by systemically administered LPS, but no apparent effect of enoxaparin. However, Authors observed significant drop in LPS-induced neurofilament light chain (intermediate filament specific for neurons) release in enoxaparin treated animals, suggesting that inhibition of coagulation cascade is indeed neuroprotective. Moreover, LPS triggered TNFa mRNA expression in hippocampus, but the observed increase wasn’t inhibited by enoxaparin. On the other hand, TNFa protein level in serum was lower when enoxaparin was co-applied. Thrombin activity, as measured in whole hippocampus, was inhibited by enoxaparin. Interestingly, enoxaparin doesn’t inhibit thrombin activity in sciatic nerve and skin fragment, although one may expect its concentration and thus effect would be stronger in periphery than in brain. The interesting, but also very complex picture of interplay between systemic and local CNS coagulation and inflammation is drawn, and the presented work proves that studying effects of systemic inflammatory stimuli on neuroinflammation and coagulation in CNS is indeed very challenging.

Minor points.

  1. Line 288: “High levels of NfL measured in the serum of LPS mice may originate from the CNS, peripheral nerves, or both. However, very high levels of serum neurofilament support central neuronal damage with blood spillage [52,53]”. Do Authors claim that neurofilament release upon LPS injection is massive, thus it must be released from neurons dying in the brain? To prove this, brain damage in the described model should be more thoroughly studied, showing for instance neurons dying in the hippocampus. Since this kind of evidence is not provided, the statements like the one above need to be avoided.

We thank the reviewer; indeed, we have not evaluated a direct and exclusive damage to the brain. The comment was removed, and the text was edited accordingly.

  1. Line 192: “TNF-α was significantly increased in the LPS group compared to controls (1.57±0.18 and 1±0.05, respectively, Kruskal-Wallis, p=0.02). TNF-α expression in the enoxaparin treated LPS group was not affected by treatment and was similar to LPS and control (1.6±0.18, 1.6±0.27, p=0.99, 0.07 respectively, Figure 4E)” – here TNFa levels were similar in LPS group (1.57 or 1.6?) and enoxaparin and LPS group, but not the control. Please correct.

This was corrected “TNF-α was significantly increased in the LPS group compared to controls (1.57±0.18 and 1±0.05, respectively, Kruskal-Wallis, p=0.02). TNF-α expression in the enoxaparin treated LPS group was not affected by treatment and was similar to LPS (1.6±0.27, p=0.99, Figure 4E).” This was added to the text (page 6, lines 192-195).

  1. Was the effect of TNFa on coagulation and in reverse, effect of thrombin on TNFa expression, previously studied, especially in cells like neurons, astrocytes or microglia in any in vitro system? How do they affect each other?

We thank the reviewer for raising this important point. Indeed, thrombin was reported to activate microglia in vitro via a PAR1 mechanism, resulting in release of TNF-α. This was added to the manuscript (page 9, lines 268-272). “Thrombin was previously shown to activate C6 glioma cells in vitro, resulting in expression of TNF-α [50]. Evidence support activation of microglia via PAR1 pathway [51]. Activation of cultured microglia results in release of proinflammatory cytokines including TNF-α to the medium, which in turn reduces dopaminergic neurons [52]” .

Reviewer 2 Report (Previous Reviewer 2)

I DO NOT agree with their responses #2 and #3. I still think that the authors MUST do experiments, cardiac perfusion and reading without tissues.  These are simple experiments but will improve scientific rigors significantly.  Their responses are completely based on the possibility that their model affects brain blood vessels and having tissues in plate readers has no interference in reading.  I strongly believe that the authors SHOULD NOT assume but prove these with experimental data.  

Thanks.

Author Response

Reviewer 2

I DO NOT agree with their responses #2 and #3. I still think that the authors MUST do experiments, cardiac perfusion and reading without tissues. These are simple experiments but will improve scientific rigors significantly. Their responses are completely based on the possibility that their model affects brain blood vessels and having tissues in plate readers has no interference in reading. I strongly believe that the authors SHOULD NOT assume but prove these with experimental data.

We addressed these comments by conducting new experiments specifically designed to answer (1) whether cardiac perfusion is essential for the measurement of thrombin activity from brain tissue and (2) is there a difference or significant advantage to conduct the fluorescence reading without the tissue in the tested wells. We conducted the experiment on 10 male mice. We applied cardiac perfusion to 6 out of the 10 mice and the other 4 were not perfused. We moved the brain, dissected it to right and left hippocampus and one part of the cerebellum, and conducted the thrombin activity assay in the presence and absence of the tissue.

The results indicate that there is no significant difference between thrombin activity levels measured in the perfused tissue and the non-perfused tissue (Figure S1). We conclude that the extensive washes we conduct to the brain as part of the routine procedure is sufficient to eliminate the thrombin activity derived from the circulating blood and the cardiac perfusion holds no further advantage. This is similar to what we have found in many other tissue samples in previous studies and indicates that performing cardiac perfusion will not critically affect the results. Furthermore, in certain pathological states which affect blood vessels, performing cardiac perfusion may potentially introduce significant confounders. We therefore do not generally perform cardiac perfusion when performing thrombin activity assays.

Figure S1: Cardiac perfusion does not significantly affect the thrombin activity measurements in the mice hippocampus. Thrombin activity was measured in the hippocampus derived from mice with cardiac perfusion (N=6) and without cardiac perfusion (N=4). Results are presented as mean±SEM

Our results found thrombin activity can be measured in the absence of the tissue but holds some benefits in the presence of the tissue. Indeed, the presence of the tissue causes a consistent higher level of presumably auto-fluorescence which does not change with time and does not influence the slope of the curve resulting from enzymatic activity (Blue squares compared to pink circles in Figure S2). After the samples are removed from the buffer the slope of enzymatic activity remains steady. The presence of the sample in the buffer very rarely (about one in 100 reads) causes a fluctuation in the read which are easily excluded from the analysis. When the samples are moved to a new substrate buffer the levels of enzyme activity are much lower but still measurable (black triangles in Figure S2). We conclude from this set of experiments and from our previous experience of the thrombin activity assay that performing the assay in the presence of tissue allows us to reliably measure enzyme activity without removing the tissue and this enables us also to measure all the activity potentially generated by the assay. Removing the tissue after a set period of time is also possible but has the relative disadvantages of losing a small but measurable level of activity in the tissue itself (represented by the black triangles in Figure S2) together with limiting the assay to a certain time period which may not be optimal for measuring very high or very low levels of activity.

Figure S2: Thrombin activity measurements in the presence and absence of brain tissue in the well. 10 samples were placed in the wells and measured for 15 minutes (Pink circles until the time indicated by the horizontal arrow). Following these 5 samples were moved to new wells and the substrate added (Black triangles), measurements were continued in the well without the samples (Blue squares), and measurements were continued if the other 5 samples without removing the tissue (Pink circles) Results are presented as mean±SEM

We thank the reviewer for the opportunity to clarify these important methodological issues. We hope that the experiments performed have addressed the issues raised and validated the method for measuring thrombin activity in tissue samples. We have added a summary of these experiments to the Methods section (page 12, lines 409-411,433-437).

Round 2

Reviewer 2 Report (Previous Reviewer 2)

The extra experiments much improves the manuscript, but I still have major concern in the thrombin activity assay. It seems that measuring thrombin activity without the tissues has markedly less variations. Therefore, the authors SHOULD repeat the experiment in Figure 3 without the tissues due to the significant variations in the data. Also, I would strongly recommend incubating tissues in a tube with the buffer then removing tissues by centrifuges to obtain clearer sample for plate reading. 

This manuscript is a resubmission of an earlier submission. The following is a list of the peer review reports and author responses from that submission.

Round 1

Reviewer 1 Report

In the submitted manuscript Berkowitz and colleagues describe effects of heparin derivative, enoxaparin, on nervous system damage induced by LPS. First, LPS and enoxaparin effects on mice behavior, motoric function and learning capabilities is described, followed by measurements of thrombin activity in different tissues, including hippocampus, sciatic nerve and skin. The expression of genes important for coagulation and inflammation processes is measured in hippocampus of mice injected with LPS with or without enoxaparin. Next, neurofilament content in serum of experimental animals is assessed. Finally, the effect of enoxaparin application on the systemic inflammation triggered by LPS is evaluated.

MAJOR:

1.    The weakest point of the paper is the lack of enoxaparin effect on LPS-dependent changes in mice behavior, motoric function, and learning skills, regardless of normalization of neurofilament levels in serum obtained in the presence of enoxaparin. Authors decision on experiment timing (Figure 6) is not clear. One may expect that cognitive or motoric function deficits triggered by LPS are manifested after neuronal damage (neurofilament release) takes place. Here, tests are performed earlier, and there is no effect of enoxaparin. To substantiate the main observation, e.g. neuroprotective effect of enoxaparin, the behavioral etc. test should be applied after LPS-triggered neuronal damage is detected. 

2.    Statistics. Three groups are compared in every experiment. In case of non-normal data distribution, Kruskal-Wallis test is appropriate, and the post-test and correction type should be indicated for each experiment. In some experiments (Figure 3A, Figure 4E) LPS plus enoxaparin group is compared to control group only. Here, the difference between LPS group and LPS plus enoxaparin group need to be evaluated, like in Figure 4A.  

3.    Language is very uneven throughout the manuscript, and except for Results section requires editing.

4.    Authors mention several times that some work (treatment?) should be done on human patients. Planning clinical trial should be based on solid preclinical evidence. Are Authors convinced that the delivered data are advanced enough to substantiate switching to human subjects?

Minor:

1.       Enoxaparin should be briefly introduced in the abstract.

2.     Line 160. It is totally fine to compare CNS, large nerve and some other tissue, however calling skin “neural associated tissue” and attributing it to the same category as hippocampus is quite unfortunate. It would be better to omit this “”descriptor”” in the text. Nerve endings are present within skin, however they are vastly outnumbered by other cell types, like epithelia or fibroblasts. If someone want to study nerve endings, IHC or FISH would be more appropriate than checking skin crude extracts.

3. The Special Issue is devoted to the Development of Dopaminergic Neurons, dopaminergic system is not mentioned in the manuscript.

4. Line 200. The descriptions like “to some extent” should be avoided in scientific journals.

Author Response

Dear reviewer

We appreciate your educational comments, and we thoroughly reviewed and edited the manuscript accordingly. The multiple changes that were made can be seen in the full texts and are marked in blue. In addition to this, specific point-by-point responses are presented below.

MAJOR:

  1. The weakest point of the paper is the lack of enoxaparin effect on LPS-dependent changes in mice behavior, motoric function, and learning skills, regardless of normalization of neurofilament levels in serum obtained in the presence of enoxaparin. Authors decision on experiment timing (Figure 6)is not clear. One may expect that cognitive or motoric function deficits triggered by LPS are manifested after neuronal damage (neurofilament release) takes place. Here, tests are performed earlier, and there is no effect of enoxaparin. To substantiate the main observation, e.g. neuroprotective effect of enoxaparin, the behavioral etc. test should be applied after LPS-triggered neuronal damage is detected. 

Previous studies have shown that sickness behavior appears immediately following LPS injection and returns to baseline after 24 hours (Perez-Dominguez 2019 Neural Regen Res). Cognitive changes are detected in the first days following injection and improve over 7 days (Zhao et. al, Scientific Reports 2019). Therefore, there is only a narrow time window for the conduction of meaningful behavioral cognitive tests when there is a relative absence of sickness behavior. This was added to the text Introduction (page 2 lines 92) and Discussion (page 8 lines 249-252).

The SIMOA is a highly sensitive assay used to measure serum levels of NfL, and an elevation can be detected at the beginning of neuronal damage, preceding changes in behavior and motor abilities (Khalil et. al, Nature Reviews Neurology 2018). Thus, it is reasonable to assume that elevated NfL will also be found in the absence of cognitive deficits. Behavioral cognitive deficits are multifactorial and thus more difficult to assess. Our findings are in line with previous works and are strengthened by the the major statements of this study which are the coagulation and neuronal involvement in the LPS model rather than the cognitive induced effects.

  1. Three groups are compared in every experiment. In case of non-normal data distribution, Kruskal-Wallis test is appropriate, and the post-test and correction type should be indicated for each experiment. In some experiments (Figure 3A, Figure 4E) LPS plus enoxaparin group is compared to control group only. Here, the difference between LPS group and LPS plus enoxaparin group need to be evaluated, like in Figure 4A.  

We thank the reviewer for this comment and edited the text accordingly (pages 5, lines 167-169 and page 6, lines 195-197). The two major comparisons of interest were initially defined between the healthy control animals and the LPS injected animals and between the LPS injected treated and untreated groups. Thus two independent comparisons were applied.  

  1. Language is very uneven throughout the manuscript, and except for Results section requires editing.

We have edited the manuscript per the reviewer’s suggestion.

  1. Authors mention several times that some work (treatment?) should be done on human patients. Planning clinical trial should be based on solid preclinical evidence. Are Authors convinced that the delivered data are advanced enough to substantiate switching to human subjects?

Of course, we agree with the reviewer that our novel treatment should be further evaluated by preclinical studies. However, observational studies on the effect of enoxaparin (given as part of a commonly indicated treatment in bedridden patients) during infections are needed. This clarification was added to the discussion section (page 9, lines 261-263)

Minor:

  1. Enoxaparin should be briefly introduced in the abstract. This was corrected (page 1 line 20)
  2. Line 160. It is totally fine to compare CNS, large nerve and some other tissue, however calling skin “neural associated tissue” and attributing it to the same category as hippocampus is quite unfortunate. It would be better to omit this “”descriptor”” in the text. Nerve endings are present within skin, however they are vastly outnumbered by other cell types, like epithelia or fibroblasts. If someone want to study nerve endings, IHC or FISH would be more appropriate than checking skin crude extracts.

We agree with the reviewer that the comparison between tissues that are clearly neuronal, and the skin is confusing. Recently, a report by our group (Golderman et, al, Biomedicines, 2022) found a significant correlation between skin innervation and skin thrombin activity levels. That is the rationale for using a skin sample for studying the effects of inflammation of the thrombin pathway. This was added to the Methods section (page 5 line 162-163).

  1. The Special Issue is devoted to the Development of Dopaminergic Neurons, dopaminergic system is not mentioned in the manuscript.

Indeed, this manuscript was submitted to the section of Molecular neurobiology, and not specifically to the special issue.

  1. Line 200. The descriptions like “to some extent” should be avoided in scientific journals.

This was corrected

Reviewer 2 Report

The study tries to show that LPS-induced inflammation increases hippocampal thrombin activity and PAR mRNA levels to induce coagulation, which is correlated with elevation of neuroinflammation factors, including NfL and TNF-alpha, and altered behaviors. Moreover, these pathological changes are reversed by enoxaparin, an anticoagulant. Thus, neuroinflammation-induced coagulation and neuronal damages can be reversed by enoxaparin. Although the concept is highly intriguing, the hippocampus thrombin activity assay has a serious flaw.

Ex vivo experiments are not physiological, and as a result, do not accurately represent coagulation in vivo, as the prior study has already demonstrated (Hofer et al., J. Thromb. Haemostasis 2019). In fact, research have previously created in vivo thrombin activity probes (Wang et al., ACS Appl. Mater. Interfaces. 2021 and Chen et al., J. Neurosci. 2012). In order to accurately assess thrombin activity in the hippocampus, a fluorometric assay using the synthetic peptide substrate in a 96-well format is not appropriate. Authors also said that measurements were made using a microplate reader after placing whole hippocampus tissues in a 96-well microplate without any processing. This unlikely gives accurate reading. Taken together, authors must use an appropriate in vivo assays to measure hippocampal thrombin activity.

There is no rationale for excluding female animals.

There are several overstatements in the manuscript. For example, authors have not shown neuronal structure and function, which are different from behaviors. Also, in method 4.8, authors have not measured protein levels. These statement should be revised.

Author Response

Dear reviewer

We appreciate your educational comments, and we thoroughly reviewed and edited the manuscript accordingly. The multiple changes that were made can be seen in the full texts and are marked in blue. In addition to this, specific point-by-point responses are presented below.

Ex vivo experiments are not physiological, and as a result, do not accurately represent coagulation in vivo, as the prior study has already demonstrated (Hofer et al., J. Thromb. Haemostasis 2019). In fact, research have previously created in vivo thrombin activity probes (Wang et al., ACS Appl. Mater. Interfaces. 2021 and Chen et al., J. Neurosci. 2012). In order to accurately assess thrombin activity in the hippocampus, a fluorometric assay using the synthetic peptide substrate in a 96-well format is not appropriate. Authors also said that measurements were made using a microplate reader after placing whole hippocampus tissues in a 96-well microplate without any processing. This unlikely gives accurate reading. Taken together, authors must use an appropriate in vivo assays to measure hippocampal thrombin activity.

Indeed, using the in-vivo method may add another tier of information regarding the coagulation involvement in this model, however, it is dependent on the permeability of the BBB which may be questionable in this model, and emphasizes BBB disruption rather then intrinsic brain coagulation-derived activity. This probably is not relevant to our experimental setting and was beyond the scope of this manuscript. This was added to the Discussion section (page 10 line 334-335).  

 Ex-vivo experiments are not physiological, however, they allow for the measurement of the low levels of cell-derived thrombin, and specifically for thrombin activity generated and secreted by neuronal cells. This measurement method reduces the confounding effect of serum-derived thrombin activity. We and others have shown the use of this well-established method in several neurological models. This assay is applied on whole tissue and not on tissue-homogenates preparation and was previously found by electrophysiological experiment to be structure and function dependent (changed in response to depolarization) thus we believe this method represents the in-vivo physiological response of the tissue.  

Furthermore, our measurements were compared to each other, thus indicating the effect of LPS and enoxaparin.

There is no rationale for excluding female animals.

Including females is associated with increased variability especially in the behavioral tests due to the estrous cycle, thus it would necessitate repeating all the experiments during different time points of the ovulation cycle, to exclude a hormonal effect on measurements. This would have resulted in a high number of animals in each group. Due to ethical considerations, experiments were conducted on males only, to allow us to keep the number of animals used as low as possible. After finetuning the effects reported in the present study, further evaluation of more specific interventions in females will be conducted.  

There are several overstatements in the manuscript. For example, authors have not shown neuronal structure and function, which are different from behaviors.

We agree with the reviewer, this was corrected (page 1, line 32)

Also, in method 4.8, authors have not measured protein levels. These statement should be revised.

We thank the reviewer for pointing this out and have corrected the title accordingly.

Round 2

Reviewer 1 Report

All major points have been adequately addressed.

Only one small issue: sometimes Authors refer to LPS treated animals as "untreated LPS" (line 165 and others), and it's quite confusing for the reader. Changing the treatment description to "LPS" or "LPS only" would help to remove this confusion. 

Author Response

We thank the reviewer and edited the text accordingly

Reviewer 2 Report

1.     The authors comment that their measurement of thrombin activity is derived from neuronal cells not from serum. This makes the story much more complicated. If the authors examine neuron-derived thrombin instead of serum-derived thrombin, authors MUST perform cardiac perfusion to remove all blood before their measurement to avoid potential contamination. Also, this information MUST be included in the manuscript.

2.     The authors’ measurement is still questionable. The authors comment that they used whole tissue in the 96-well setup. How did authors control samples? The authors mentioned that they used the left half of the hippocampus for measurement. Did they use the same tissue for the protein assay? or use the right half? In addition, can the substrate the authors used penetrate entire hippocampal tissues? How deep can the plate reader read? Without this information, I assume that the measurement seems only superficial layer of the tissue, which does not represent in vivo physiology.

3.     Authors also mention the validation of their assay, including electrophysiology and structure and functional assays. How does this validation relate in vivo physiology? This information MUST be presented. 

Author Response

Dear Reviewer,

Thank you for your thorough and positive review of the manuscript titled “LPS Induced Coagulation and Neuronal Damage in a Mice Model is Attenuated by Enoxaparin” (ijms-1807824). We appreciate your positive feedback. Below is our point-by-point response to each of the comments in blue.

Thank you for this opportunity and your support,

Efrat Shavit-Stein, on behalf of the authors

Comments and Suggestions for Authors

  1. The authors comment that their measurement of thrombin activity is derived from neuronal cells not from serum. This makes the story much more complicated. If the authors examine neuron-derived thrombin instead of serum-derived thrombin, authors MUST perform cardiac perfusion to remove all blood before their measurement to avoid potential contamination. Also, this information MUST be included in the manuscript.

The thrombin assay, which is one of the methods used in the present study, is not new. We have developed the thrombin activity assay to measure ex-vivo brain tissue over the past 20 years and this work has been described in over 30 publications many of which are cited below. Indeed, as the reviewer has commented, there were a number of issues to be addressed in setting up this assay. In the first publications of Beilin et al. we found that using brain homogenates was inappropriate due to the release of large amounts of thrombin inhibitors. We established a highly sensitive assay for neuronal cell culture and then found that this assay produced reliable results in brain slices and isolated brain regions. We found that using whole tissue enabled the reliable measurement of thrombin activity in tissue slices from the brain. The issue of brain perfusion was addressed in depth in the studies of Bushi et al on animals with stroke. Perfusion of animals (cardiac perfusion) to eliminate blood from brain vessels in the tissue has limitations. In many of these neurological-involved models, increased thrombosis may lead to vascular occlusions and applying perfusion under these conditions may result in a biased clearance of the blood over the different parts of the tissue. In various pathologies, such as ischemic stroke, trauma and brain tumors, the effects of perfusion are inherently different in the pathological and normal tissue. Therefore, we performed the experiments on non-perfused brain slices. As the reviewer has pointed out, the thrombin activity measured inside the brain tissue probably originates from both the blood and from coagulation pathway activity intrinsic to glia in the brain. We have described the use of this method in several animal models for central and peripheral nervous system diseases such as GBM, GBS, LPS-induced neuroinflammation, epilepsy and stroke and found a reliable and reproducible signal linked to the underlying pathology. It is important to note that in inflammatory diseases increased thrombin activity measured has been demonstrated by a number of other methods (Proteomic analysis of active multiple sclerosis lesions reveals therapeutic targets, Han et al. Nature 2008), further supporting the validity of the findings based on the thrombin assay in the present manuscript. We feel that the thrombin assay for ex vivo brain tissue is well established, validated and similar methods have been used by others (Friedman et al in spinal cord and optic nerve, Wang et al on brains following global ischemia and Han et al. in brains following inflammation). We have added selected references supporting the method in the Methods section. We here include a more complete list of publications using the thrombin activity measurement assay for the benefit of the reviewer:

  1. Goldberg Z, Sher I, Qassim L, Chapman J, Rotenstreich Y, Shavit-Stein E. Intrinsic Expression of Coagulation Factors and Protease Activated Receptor 1 (PAR1) in Photoreceptors and Inner Retinal Layers. Int J Mol Sci. 2022;23(2):1–12.
  2. Shavit-Stein E, Mindel E, Gofrit SG, Chapman J, Maggio N. Ischemic stroke in PAR1 KO mice: Decreased brain plasmin and thrombin activity along with decreased infarct volume. PLoS One. 2021 Mar;16(3 March).
  3. Golderman V, Gofrit SG, Maggio N, Gera O, Gerasimov A, Laks D et al.. A novel highly sensitive method for measuring inflammatory neural-derived apc activity in glial cell lines, mouse brain and human csf. Int J Mol Sci [Internet]. 2020 Mar 31;21(7):2422.
  4. Shavit-Stein E, Rahal IA, Bushi D, Gera O, Sharon R, Gofrit SG et al. Brain protease activated receptor 1 pathway: A therapeutic target in the superoxide dismutase 1 (SOD1) mouse model of amyotrophic lateral sclerosis. Int J Mol Sci. 2020 May;21(10).
  5. Golderman V, Shavit-Stein E, Gera O, Chapman J, Eisenkraft A, Maggio N. Thrombin and the Protease-Activated Receaptor-1 in Organophosphate-Induced Status Epilepticus. J Mol Neurosci. 2019 Feb 4;67(2):227–34.
  6. Gera O, Shavit-Stein E, Chapman J. The Effect of Neuronal Activity on Glial Thrombin Generation. J Mol Neurosci. 2019 Apr 25;67(4):589–94.
  7. Shavit-Stein E, Sheinberg E, Golderman V, Sharabi S, Wohl A, Gofrit SG et al. A Novel Compound Targeting Protease Receptor 1 Activators for the Treatment of Glioblastoma. Front Neurol. 2018 Dec 17;9:1087.
  8. Shavit Stein E, Ben Shimon M, Artan Furman A, Golderman V, Chapman J, Maggio N. Thrombin Inhibition Reduces the Expression of Brain Inflammation Markers upon Systemic LPS Treatment. Neural Plast. 2018 Jun 19;2018.
  9. Bushi D, Chapman J, Wohl A, Stein ES, Feingold E, Tanne D. Apixaban decreases brain thrombin activity in a male mouse model of acute ischemic stroke. J Neurosci Res [Internet]. 2018 Aug 1;96(8):1406–11.
  10. Friedmann I, Faber-Elman A, Yoles E, Schwartz M. Injury-induced gelatinase and thrombin-like activities in regenerating and nonregenerating nervous systems. FASEB J [Internet]. 2018 Mar;13(3):533–43.
  11. Ben Shimon M, Zeimer T, Shavit Stein E, Artan-Furman A, Harnof S, Chapman J et al. Recovery from trauma induced amnesia correlates with normalization of thrombin activity in the mouse hippocampus. Ai J, editor. PLoS One. 2017 Nov 28
  12. Reuveni G, Golderman V, Shavit-Stein E, Rosman Y, Shrot S, Chapman J et al. Measuring thrombin activity in frozen brain tissue. Neuroreport. 2017;28(17):1176–9.
  13. Bushi D, Stein ES, Golderman V, Feingold E, Gera O, Chapman J et al. A linear temporal increase in thrombin activity and loss of its receptor in mouse brain following ischemic stroke. Front Neurol. 2017 Apr 10;8(APR):138.
  14. Itsekson-Hayosh Z, Shavit-Stein E, Katzav A, Rubovitch V, Maggio N, Chapman J et al. Minimal Traumatic Brain Injury in Mice: Protease-Activated Receptor 1 and Thrombin-Related Changes. J Neurotrauma. 2016 Oct 15;33(20):1848–54.
  15. Golderman V, Shavit-Stein E, Tamarin I, Rosman Y, Shrot S, Rosenberg N et al. The Organophosphate paraoxon and its antidote obidoxime inhibit thrombin activity and affect coagulation in vitro. PLoS One. 2016 Sep 1;11(9).
  16. Deselms H, Maggio N, Rubovitch V, Chapman J, Schreiber S, Tweedie D et al.. Novel pharmaceutical treatments for minimal traumatic brain injury and evaluation of animal models and methodologies supporting their development. J Neurosci Methods. 2016 Oct 15;272:69–76.
  17. Stein ES, Itsekson-Hayosh Z, Aronovich A, Reisner Y, Bushi D, Pick CG et al. Thrombin induces ischemic LTP (iLTP): implications for synaptic plasticity in the acute phase of ischemic stroke. Sci Rep. 2015 Jul 21;5(1):7912.
  18. Shimon M Ben, Lenz M, Ikenberg B, Becker D, Stein ES, Chapman J et al. Thrombin regulation of synaptic transmission and plasticity: Implications for health and disease. Front Cell Neurosci. 2015 Apr 21 9(APR):151.
  19. Itsekson-Hayosh Z, Shavit-Stein E, Last D, Goez D, Daniels D, Bushi D et al. Thrombin Activity and Thrombin Receptor in Rat Glioblastoma Model: Possible Markers and Targets for Intervention? J Mol Neurosci. 2015;56(3):644–51.
  20. Bushi D, Ben Shimon M, Shavit Stein E, Chapman J, Maggio N, Tanne D. Increased thrombin activity following reperfusion after ischemic stroke alters synaptic transmission in the hippocampus. J Neurochem. 2015;135(6):1140–8.
  21. Maggio N, Itsekson Z, Ikenberg B, Strehl A, Vlachos A, Blatt I et al. The anticoagulant activated protein C (aPC) promotes metaplasticity in the hippocampus through an EPCR-PAR1-S1P1 receptors dependent mechanism. Hippocampus. 2014 Aug;24(8):1030–8.
  22. Itzekson Z, Maggio N, Milman A, Shavit E, Pick CG, Chapman J. Reversal of trauma-induced amnesia in mice by a thrombin receptor antagonist. J Mol Neurosci. 2014 Dec 19;53(1):87–95.
  23. Bushi D, Chapman J, Katzav A, Shavit-Stein E, Molshatzki N, Maggio N et al. Quantitative detection of thrombin activity in an ischemic stroke model. J Mol Neurosci. 2013;51(3):844–50.
  24. Maggio N, Itsekson Z, Dominissini D, Blatt I, Amariglio N, Rechavi G et al. Thrombin regulation of synaptic plasticity: implications for physiology and pathology. Exp Neurol 2013;247:595–604.
  25. Maggio N, Cavaliere C, Papa M, Blatt I, Chapman J, Segal M. Thrombin regulation of synaptic transmission: Implications for seizure onset. Neurobiol Dis. 2013 Feb;50(1):171–8.
  26. Wang J, Jin H, Hua Y, Keep RF, Xi G. Role of protease-activated receptor-1 in brain injury after experimental global cerebral ischemia. Stroke. 2012 Sep;43(9):2476–82.
  27. Shavit E, Michaelson DM, Chapman J. Anatomical localization of protease-activated receptor-1 and protease-mediated neuroglial crosstalk on peri-synaptic astrocytic endfeet. J Neurochem. 2011;119(3):460–73.
  28. Shavit-Stein E, Beilin O, Korczyn AD, Sylantiev C, Aronovich R, Drory VE et al. Thrombin receptor PAR-1 on myelin at the node of Ranvier: A new anatomy and physiology of conduction block. Brain. 2008;131(4):1113–22.
  29. Friedmann I, Yoles E, Schwartz M. Thrombin attenuation is neuroprotective in the injured rat optic nerve. J Neurochem. 2001;76(3):641–9.
  30. Beilin O, Gurwitz D, Korczyn AD, Chapman J, Quantitative measurements of mouse brain thrombin-like and thrombin inhibition activities. Neuroreport . 2001;12(11):2347–51.
  31. Friedmann I, Hauben E, Yoles E, Kardash L, Schwartz M. T cell-mediated neuroprotection involves antithrombin activity. J Neuroimmunol. 2001 Dec 3;121(1–2):12–21.
  32. Friedmann I, Faber-Elman A, Yoles E, Schwartz M. Injury-induced gelatinase and thrombin-like activities in regenerating and nonregenerating nervous systems. FASEB J  Off Publ Fed Am Soc  Exp Biol. 1999 Mar;13(3):533–43.
  33. Bushi D, Gera O, Kostenich G, Shavit-Stein E, ‏ Bushi D, Gera O, Kostenich G, Shavit-Stein E, Neuroscience 2016‏  A novel histochemical method for the visualization of thrombin activity in the nervous system‏.

  1. The authors’ measurement is still questionable. The authors comment that they used whole tissue in the 96-well setup. How did authors control samples? The authors mentioned that they used the left half of the hippocampus for measurement. Did they use the same tissue for the protein assay? or use the right half? In addition, can the substrate the authors used penetrate entire hippocampal tissues? How deep can the plate reader read? Without this information, I assume that the measurement seems only superficial layer of the tissue, which does not represent in vivo physiology.

Indeed, we used one side of the hippocampus for the thrombin activity measurement followed by the protein assay (the right side) and the other (left side) was used for mRNA purification and analysis of gene expression. One side of the hippocampus of adult mice fits easily into the well of a 96-well plate. The tissue is placed in a well containing a liquid buffer. A liquid fluorogenic substrate is added and the measurement is conducted continuously at 37°C while regularly shaking the plate. The thrombin is secreted from the tissue to the surrounding buffer, accumulates and by simple diffusion reaches a steady state. The highly sensitive Infinite F Nano+ by Tecan instrument using the Top-reading mode allows to conduct fluorescence measures from the well-containing well (both excitation and emission applies vertically) in high sampling rate and 25 flashes per read and 25µs integration time. The activity measured is generated both by the small substrate (MW=770) entering the small fragment of brain tissue and being converted there to its active form by thrombin and also by an amount of soluble thrombin diffusing out of the tissue. Thrombin activity is to a large amount generated in-vivo and preserved in the ex-vivo tissue. As detailed in the following point, the ex-vivo tissue does retain some capacity to generate thrombin in response to physiological challenges such as depolarization (Shavit et al. Anatomical localization of protease-activated receptor-1 and protease-mediated neuroglial crosstalk on peri-synaptic astrocytic endfeet. J Neurochem. 2011).     

  1. Authors also mention the validation of their assay, including electrophysiology and structure and functional assays. How does this validation relate in vivo physiology? This information MUST be presented. 

As detailed above, we have developed the ex-vivo thrombin activity that can be used both on brain slices and in neuronal preparations and cell culture. A number of our studies have indicated that the thrombin activity measured can be modulated physiologically in the ex-vivo tissue: In a brain slice preparation, we have shown that oxygen-glucose deprivation (OGD) increases thrombin activity (Stein Efrat Shavit et al. Thrombin induces ischemic LTP (iLTP): implications for synaptic plasticity in the acute phase of ischemic stroke. Sci Rep. 2015). In a brain synaptosome preparation we have found that depolarization induces thrombin secretion (Shavit et al. Anatomical localization of protease-activated receptor-1 and protease-mediated neuroglial crosstalk on peri-synaptic astrocytic endfeet. J Neurochem. 2011). In a Schwann cell culture, we found that depolarization significantly modulates thrombin secreted from the cells in a time-dependent manner (Gera et al, The Effect of Neuronal Activity on Glial Thrombin Generation, J Mol Neurosci. 2019). In a recent study we have found that in isolated mouse retina thrombin activity is significantly modulated by depolarization (Goldberg et al, Intrinsic Expression of Coagulation Factors and Protease Activated Receptor 1 (PAR1) in Photoreceptors and Inner Retinal Layers, Int J Mol Sci . 2022).

In the present study we did not manipulate the ex-vivo tissue samples physiologically and therefore assume that the excess thrombin measured represents inflammation induced in-vivo and subsequently rapidly measured ex-vivo.
